# *Lactiplantibacillus plantarum* 1008 Promotes Reproductive Function and Cognitive Activity in Aged Male Mice with High-Fat-Diet-Induced Obesity by Altering Metabolic Parameters and Alleviating Testicular Oxidative Damage, Inflammation and Apoptosis

**DOI:** 10.3390/antiox13121498

**Published:** 2024-12-09

**Authors:** Chin-Yu Liu, Tsung-Yu Tsai, Te-Hua Liu, Ting-Chia Chang, Yi-Wen Chen, Chih-Wei Tsao

**Affiliations:** 1Department of Nutritional Science, Fu Jen Catholic University, New Taipei City 242062, Taiwan; nf351.lab@gmail.com (C.-Y.L.); ctc5628@gmail.com (T.-C.C.); angela55106577@gmail.com (Y.-W.C.); 2Department of Food Science, Fu Jen Catholic University, New Taipei City 242062, Taiwan; tytsai@mail.fju.edu.tw (T.-Y.T.); r19901217@hotmail.com (T.-H.L.); 3Division of Urology, Department of Surgery, Tri-Service General Hospital, National Defense Medical Center, Taipei 11490, Taiwan

**Keywords:** aged, obesity, hypogonadism, memory, spermatogenesis, testosterone

## Abstract

The effects of *Lactiplantibacillus plantarum* 1008 (LP1008) on age-related cognitive impairment and skeletal muscle atrophy have been reported previously. However, its role in obesity- and age-related hypogonadism has yet to be explored. This study investigates the therapeutic efficacy of low- and high-dose LP1008 in a high-fat-diet-fed male mouse model. Mice at 37 weeks of age were fed a standard diet (*n* = 8) or a 45% high-fat diet for 28 weeks, and the high-fat-diet-fed mice were divided into vehicle, low-dose and high-dose LP1008 groups (*n* = 8 per group) on the basis of the treatment administered for an additional 8 weeks. We found that LP1008 suppressed the increases in total cholesterol levels and liver function parameters and alleviated histological changes in the brain, ileum, gastrocnemius muscle and testes. In terms of reproductive function, LP1008 attenuated the decreases in sperm quality, sperm maturity, testosterone levels and levels of enzymes involved in testosterone biosynthesis. Furthermore, LP1008 altered impairments in spatial learning and memory and induced slight alterations in the gut microbiota. Moreover, LP1008 exerted antioxidant, anti-inflammatory and anti-apoptotic effects in aged, obese male mice. LP1008 reversed diet-induced obesity, age-related reproductive dysfunction and pathological damage by increasing testosterone levels and altering the gut microbiome through the regulation of mediators involved in oxidative stress, apoptosis and inflammation.

## 1. Introduction

Due to the increased lifespan and rapidly aging population, there is growing concern regarding age-related health problems, including late-onset hypogonadism (LOH) [1,2]. LOH is characterized by a progressive decline in testosterone levels, mainly beginning in middle life, and is associated with various symptoms, such as erectile dysfunction, loss of libido, diminished morning erections, hot flashes, decreased muscle mass and strength, increased body fat, memory decline, concentration impairment and depression [3,4]. The European Male Aging Study (EMAS) suggested that LOH should be defined as either a total testosterone level <8 nmol/L or a total testosterone level between 8 and 11 nmol/L with a free testosterone level <220 pmol/L, combined with symptoms of diminished morning erections, erectile dysfunction and loss of libido [5]. Furthermore, a prospective study adopting data from the EMAS indicated that, after controlling for factors such as age, body mass index, and poor general health, men with severe LOH had a five-fold increased risk of all-cause mortality when compared with men without LOH. Men with testosterone levels <8 nmol/L (without symptoms) and those with three sexual symptoms (without low testosterone levels) had two- and three-fold greater risks of mortality, respectively [6].

It has been reported that 2–6% of men aged 40–79 years suffer from LOH [7]; however, in many cases, LOH is thought to be unrecognized due to its association with various non-specific syndromes that are considered unavoidable because of aging, or are overlooked by individuals with other comorbidities [8,9]. A cross-sectional study revealed that men aged 40 years and older with hypertension, dyslipidemia, obesity, metabolic syndrome, chronic obstructive pulmonary disease and diabetes have a greater risk of developing LOH [10]. Obesity is one of the most significant modifiable risk factors for LOH; in particular, obesity increases the risk of LOH via the suppression of the hypothalamic–pituitary–gonadal (HPG) axis mediated by the dysregulation of inflammatory cytokines [11]. Considering the close association between obesity-related hypogonadism and systemic inflammation, the theory (gut endotoxins leading to a decline in gonadal function) proposed by Tremellen in 2016 suggested a potential mechanism involving the release of bacterial lipopolysaccharide (LPS), the gut microbiota and obesity-related low testosterone levels [12]. Briefly, gut dysbiosis caused by obesity and a high-fat diet promotes endotoxin release, as LPS is released into the circulation and initiates systemic inflammation, thereby diminishing testosterone production [11,12]. *Lactiplantibacillus plantarum*, previously known as *Lactobacillus plantarum*, is a widely used probiotic that has been shown to promote health through preventing and treating intestinal inflammation and metabolic abnormalities [13,14]. The novel probiotic *L. plantarum* 1008 (LP1008), which was isolated from Taiwanese pickled cabbage, has been shown to improve age-related functional findings, including muscle mass and strength and cognitive impairment, in animal [15] and human [16] studies. However, its role in obesity- and age-related hypogonadism has yet to be explored. Therefore, this study aimed to establish a high-fat-diet-induced aged mouse model and investigate the effects of low and high doses of LP1008 on obesity, age-related biochemical parameters, pathological changes, spatial learning and memory, the gut microbiome, reproductive function and the underlying mechanisms.

## 2. Materials and Methods

### 2.1. Animals and Treatment

Thirty-two 36-week-old male C57BL/6 mice were purchased from the National Laboratory Animal Center (Taipei, Taiwan) and acclimatized to the Laboratory Animal Center of the National Defense Medical Center (Taipei, Taiwan). The animals were housed in polycarbonate cages (4 per cage) and kept in a well-controlled environment (temperature at 21–23 °C, humidity within 50–60%, and a 12 h/12 h light/dark cycle) with free access to food and water.

LP1008 was obtained from fermented Taiwanese cabbage. The LP1008 dose (4.1 × 10^9^ CFU/kg) for the mice was determined by converting the dose used in a previous clinical trial in elderly individuals [16], on the basis of differences in body surface area between humans and mice. After 1 week of adaptation, the mice were randomly divided into groups receiving either a standard control diet (CON group; 5001, LabDiet, Richmond, IN, USA) or a high-fat diet (HFD group; D12451, Research Diets, New Brunswick, Canada) for 28 weeks, along with a vehicle control (same volume of solvent solution), low-dose LP1008 (HPL group, 4.1 × 10^9^ CFU/kg) or high-dose LP1008 (HPH group, 2.05 × 10^10^ CFU/kg) treatment by oral gavage for 8 weeks (*n* = 8 per group) (Appendix A). The compositions of the diets used in the CON and HFD groups are listed in Table 1. The mice were euthanized at 74 weeks of age, and blood, brain, liver, kidney, ileum, gastrocnemius muscle, testis, epididymis, vas deferens, epididymal fat mass and sperm samples were collected for further analyses.

### 2.2. Serum Parameters

Blood samples were collected carefully from the left ventricle and aspirated into a 1.5 mL Eppendorf tube slowly to prevent hemolysis. The collected blood was centrifuged for 10 min at 3000× *g* and 4 °C to separate the serum. Serum glucose, total cholesterol (TC), aspartate transaminase (AST) and alanine transaminase (ALT) levels were measured using a Hitachi 7180 biochemistry automatic analyzer. Serum insulin (10-1247-01; Mercodia, Uppsala, Sweden), testosterone (MBS163127; MyBioSource, San Diego, CA, USA) and LPS (A39552; Thermo Fisher Scientific, Rockford, IL, USA) levels were measured using commercially available enzyme-linked immunosorbent assay (ELISA) kits.

### 2.3. Sperm Quality

Spermatozoa were collected from the vas deferens using 500 μL of pre-warmed phosphate-buffered saline (PBS) to allow mature sperm to swim. The percentage of sperm motility was determined using a diluted sperm suspension in a counting chamber (3900; Hausser Scientific, Horsham, PA, USA) under a light microscope (DM1000; Leica, Wetzlar, Germany). The sperm count was measured using an automated cell counter (TC20; Bio-Rad, Taipei, Taiwan). In addition, drops of the sperm suspension were placed on a slide, air-dried, fixed with methanol and stained with eosin solution. The samples were examined for sperm deformity, and the deformity rate was determined for at least 250 sperm from each sample.

### 2.4. Hematoxylin–Eosin (HE) Staining

The brain, liver, kidney, ileum, gastrocnemius and testis sections were fixed with 4% formaldehyde solution at room temperature for at least 24 h and transferred to Biotools (New Taipei City, Taiwan) for tissue processing, paraffin embedding, sectioning at 4 μm with a rotary microtome (RM2125 RTS; Leica) and staining with hematoxylin and eosin (H&E). The stained slides were dehydrated, then mounted with mounting medium (No. 20402; Muto Pure Chemical, Tokyo, Japan). Images of the H&E-stained tissue sections were captured using a light microscope (DM1000, Leica) and the SPOT Imaging Software v4.6 (Sterling Heights, MI, USA).

### 2.5. Measurements of Spatial Learning and Memory

Spatial learning and memory were assessed using a Morris water maze (MWM) test [17], and monitored and recorded using an Ethovision XT tracking system (Noldus Information Technology, Leesburg, VA, USA). Briefly, a white round pool measuring 100 cm in diameter and 30 cm in height was filled with water at 25–27 °C, and the pool was divided into four quadrants (I, II, III, IV). During spatial acquisition training on 3 consecutive days (days 1–3), an escape platform with a diameter of 10 cm was placed in a fixed quadrant (III) submerged 1.5 cm below the water surface, and the mice were subjected to five trials per day. During each trial, the mice were placed facing the wall of the pool in the other three quadrants (I, II and IV) and had to find the escape platform within 90 s. If the mouse failed to find the platform within 90 s, it was guided to the platform and allowed to remain there for 30 s. The daily escape latency was recorded. A probe trial was conducted 24 h after the last day of spatial acquisition training (day 4). The escape platform was removed, and the mice were allowed to swim freely for 90 s. The percentage of time spent in the target quadrant (III) and along the pathway was recorded. To examine spatial working memory after the probe trial on 3 consecutive days (days 5–7), the platform was moved every day to the middle of the other quadrants (I, II and V), and the mice were subjected to five trials per day. The escape latency was recorded daily.

### 2.6. Measurements of Testicular Superoxide Dismutase, Catalase, Glutathione Peroxidase and Thiobarbituric Acid-Reactive Substances

Testicular superoxide dismutase (SOD; No. 706002, Cayman, MI, USA), catalase (CAT; No. 707002, Cayman), glutathione peroxidase (GPx; No. 703102) and thiobarbituric acid-reactive substances (TBARS; No. 10009055) were detected in testicular lysates using commercially available ELISA kits. Testicular lysates were extracted from testicular tissue with a radioimmunoprecipitation assay (RIPA) buffer (89900; Thermo Fisher Scientific) and a protease and phosphatase inhibitor cocktail (78440; Thermo Fisher Scientific) at a volume-to-weight ratio of 1:4, then centrifuged at 4 °C and 14,000 rpm for 15 min.

### 2.7. Western Blotting

Western blotting was initially performed with quantified testicular lysates using a detergent-compatible colorimetric protein assay kit (5000112; Bio-Rad). The lysates were separated using equal amounts of protein via 10–12% sodium dodecyl sulfate (SDS)-polyacrylamide gel electrophoresis, transferred onto polyvinylidene difluoride (PVDF) membranes, blocked in 5% nonfat dry milk (1706404; Bio-Rad) buffer and then incubated with primary antibodies overnight at 4 °C and secondary antibodies for 1 h. Finally, the signals were detected with a SPOT Xplorer camera (Diagnostic Instruments, Kentwood, MI, USA) and Clarity Max-enhanced chemiluminescence (ECL) substrate (1705062, Bio-Rad). The results were analyzed using the ImageJ software v1.53t (National Institutes of Health, Bethesda, MD, USA). Antibodies targeting StAR (sc-25806; Santa Cruz Biotechnology, Dallas, TX, USA), CYP11A1 (sc-292456), 3β-HSD (sc-28206), CYP17A1 (sc-66850), 17β-HSD (sc-135044), NF-κB (E381; Abcam, Waltham, MA, USA), TNF-α (ab1793; Abcam), IL-6 (sc-57315), Bax (#2772; Cell Signaling Technology, Beverly, MA, USA), Bcl-xl (ab32370; Abcam), Caspase 9 (#9508), Caspase 3 (#9664), PARP (#3542), Caspase 8 (GTX59607; GeneTex, California, USA), β-actin (A5316; Sigma, Saint Luis, MO, USA), GAPDH (sc-32233), goat anti-rabbit IgG-HRP (sc-2004) and goat anti-mouse IgG-HRP (sc-2005) were used.

### 2.8. Fecal Microbial Profiling

Fresh fecal samples were collected after 8 weeks of LP1008 treatment and frozen before extraction. Fecal microbiota DNA was extracted using the QIAamp PowerFecal Pro DNA Kit (Qiagen, Germantown, MD, USA) following the manufacturer’s instructions. The fecal microbiota sequencing and analysis was commissioned to Biotools (New Taipei City, Taiwan). The V3–V4 hypervariable regions of 16S ribosomal ribonucleic acid (rRNA) were amplified, purified and denoised with DADA2. Amplicon sequence variants (ASVs) were produced, and the ASV sequences were sequenced to determine the microbial composition and function.

### 2.9. Fecal Short-Chain Fatty Acids

Frozen fecal samples were extracted, and the supernatants were analyzed by a gas chromatograph (436 GC; Bruker Daltonics, Billerica, MA, USA) coupled to a triple quadrupole mass spectrometer (EVOQ GC-TQ; Bruker Daltonics) on a VF-5ms column, with helium as the carrier gas at a flow rate of 1.0 mL/min. The initial oven temperature was 40 °C for 5 min, after which the temperature was increased to 310 °C at 10 °C/min and maintained for 5 min. The inlet, transfer line, and ion source temperatures were 260, 280, and 250 °C, respectively. The solvent delay was 5 min, and the electron energy was 70 eV. The experimental data were collected using the MSWS software 8.2 (Bruker Daltonics).

### 2.10. Statistical Analysis

All statistical analyses were performed using the GraphPad Prism software 9.3.1 (GraphPad Software, San Diego, CA, USA). Descriptive statistics were used to calculate mean values and standard deviations. Student’s *t*-test was used for comparisons between two groups (the CON and HFD groups). One-way analysis of variance with post hoc Fisher’s least significant difference test was used for comparisons among multiple groups (the HFD, HPL and HPH groups) that were normally distributed, whereas the Kruskal–Wallis test with Dunn’s post hoc-test was used for comparisons among multiple groups that were not normally distributed.

## 3. Results

### 3.1. Effects of LP1008 on Body Weight, Liver Weight, Liver Histology and Biochemical Parameters in Aged and Obese Mice

After 36 weeks of HFD feeding, the mice developed significantly greater body weights, and a significant change started with sustained HFD feeding for 4 weeks in the HFD group, compared with those in the CON group. There was no significant difference among the HFD groups at baseline (HFD: 51.69 ± 4.66, HPL: 51.87 ± 2.82, HPH: 52.34 ± 2.84), while 8 weeks of treatment with low or high doses of LP1008 had no significant weight-lowering effect (Figure 1A). A greater liver weight with significant liver steatosis and lipid accumulation in H&E-stained liver tissues was observed in the HFD group, compared to the CON group. Furthermore, a significant tendency toward improving liver steatosis and decreased lipid accumulation on the basis of liver histology and no obvious change in liver weight were observed in the HPL and HPH groups (Figure 1B). Additionally, as shown in Figure 1C, the HFD group presented abnormal metabolic parameters, including increased serum glucose, insulin and TC levels. In contrast, despite similar serum glucose levels, serum insulin and TC levels were markedly improved in the HPL and HPH groups. The levels of the liver function parameters ALT and AST were significantly greater in the HFD group. Conversely, both the HPL and HPH groups presented reduced HFD-induced increases in liver enzymes (Figure 1D).

### 3.2. Effects of LP1008 on Brain Histology, Spatial Learning and Memory in Aged and Obese Mice

The histological morphology of the brain tissue sections, as determined by H&E staining, is shown in Figure 2A. The brain tissue of the HFD-fed mice was impaired, as these mice exhibited degeneration and a reduced number of neurons with unclear nucleoli, whereas the 8-week low- and high-dose LP1008 treatments resulted in the neurons becoming more closely arranged and plump with clear nucleoli. After 3 days (days 1–3) of spatial acquisition training, learning ability was observed in all groups, as demonstrated by a decrease in swimming time. The escape latency to find the escape platform in the HFD group was significantly impaired, compared with that in the CON group (Figure 2B). In contrast to the mice that were administered vehicle control, the escape latency of the mice that were administered low and high doses of LP1008 was notably shorter during the last 2 training days. The probe trial—which indicated reference memory—showed that the HFD group spent less time in the target quadrant; however, LP1008 treatment (HPL and HPH groups) significantly increased the percentage of time spent in the target quadrant compared with the HFD group (Figure 2C). Furthermore, the mice were trained for another 3 days (days 5–7) in the working memory test. Mice in the CON group found the escape platform faster than those in the HFD group throughout the training period. There was a shorter day-to-day escape latency in the HPL and HPH groups, and there was a significant difference between the LP1008 treatment groups and the vehicle control group in the last 2 days (Figure 2D).

### 3.3. Effects of LP1008 on Gastrocnemius Muscle Histology and Related Parameters in Aged and Obese Mice

As shown in Figure 3, the HFD group had a 17% lower gastrocnemius muscle mass and a 10% smaller cross-sectional area. Treatment with low and high doses of LP1008 resulted in a significantly larger area (10% and 14%, respectively) and increased gastrocnemius muscle mass.

### 3.4. Effects of LP1008 on Reproductive Organ Weight, Sperm Quality, Testicular Histology and Related Parameters in Aged and Obese Mice

HFD feeding had no effect on testis, epididymis or vas deferens weights, whereas aged and obese mice had increased epididymal fat mass regardless of LP1008 treatment (Figure 4A). In terms of sperm quality, the HFD group had significantly lower sperm motility and normal sperm morphology, with a trend toward a reduction in sperm count. While sperm motility, sperm count and normal sperm morphology were notably improved in the HPH group, the HPL group also presented significantly improved sperm motility and normal sperm morphology (Figure 4B). Testicular histological analysis revealed that damaged morphology and impaired spermatogenesis were characterized by a decreasing trend in the diameter of seminiferous tubules and thinner germinal epithelium and vacuoles, with fewer mature sperm in the HFD group than in the CON, HPL and HPH groups. Moreover, both the HPL and HPH groups presented an attenuated HFD-induced decrease in the Johnsen score, indicating improved spermatogenesis (Figure 4C).

### 3.5. Effects of LP1008 on the Levels of Serum Testosterone and Testosterone Biosynthesis-Related Proteins in Aged and Obese Mice

There were changes in the levels of serum testosterone and testosterone biosynthesis-related proteins in the HFD, HPL and HPH groups (Figure 5A). Compared with the serum testosterone levels in the NC group, those in the HFD group were suppressed 0.2-fold. Conversely, aging- and obesity-reduced serum testosterone levels were increased 4.4- and 5.5-fold after treatment with low and high doses of LP1008, respectively. Changes in testosterone biosynthesis-related protein levels were assessed, and HFD feeding resulted in significant decreases in StAR and 17β-HSD levels. However, after treatment with a low dose of LP1008, the CYP17A1 and 17β-HSD protein levels increased approximately 1.3- and 2.0-fold, respectively. After treatment with a high dose of LP1008, the StAR, CYP17A1 and 17β-HSD protein levels increased 1.4-, 1.5-, and 2.0-fold, respectively (Figure 5B).

### 3.6. Effects of LP1008 on Ileum Weight, Ileum Histology, the Gut Microbiota and Short-Chain Fatty Acid Production in Aged and Obese Mice

As shown in Figure 6A,B, the HFD group had lower ileum weights and shorter villus lengths (36% decrease) and crypt depths (22% decrease), when compared with the CON group. However, after treatment with LP1008, the length of the villi increased in both the HPL and HPH groups, where the difference in the HPH group was statistically significant. Crypt depth increased significantly in both the HPL and HPH groups. The microbial communities were analyzed after 8 weeks of LP1008 treatment. The Shannon and Simpson indices for alpha diversity showed no differences in the evenness of microbial communities among all groups, whereas the richness indices in the CON group were significantly greater than those in the HFD group. Good’s coverage in each group reached an average of 0.99, indicating good complexity of the sequencing depth in all groups (Figure 6C,D). The rank abundance curve illustrated that the HFD and HPH groups presented relatively low species abundances, while the HPL group had the highest species abundance (Figure 6E). In terms of the weighted UniFrac beta diversity of each group, the gut microbiome in the HFD group was significantly different from that in the CON group, whereas there was no difference between the LP1008-treated and HFD groups (Figure 6F). Additionally, principal coordinate analysis (PCoA) based on the weighted UniFrac distance (Figure 6G) and Bray–Curtis dissimilarity (Figure 6H) revealed that the microbial composition in the HFD group was significantly different from that in the CON group, whereas those in the HPL and HPH groups were somewhat different from that in the HFD group.

The top 10 dominant phyla in all the groups are shown in Figure 7A, from which it can be seen that *Firmicutes* (CON: 51%, HFD: 74%, HPL: 67%, HPH: 66%) and *Bacteroidetes* (CON: 47%, HFD: 24%, HPL: 29%, HPH: 31%) were the top two dominant phyla. The *Firmicutes/Bacteroidetes* ratio (F/B) was also calculated. An increased F/B ratio was observed in the obese mice, with a notably lower F/B ratio observed in the obese mice treated with low and high doses of LP1008. At the family level (Figure 7B), the top four dominant families in all groups were *Muribaculaceae* (CON: 47%, HFD: 22%, HPL: 26%, HPH: 28%), *Lachnospiraceae* (CON: 24%, HFD: 26%, HPL: 20%, HPH: 25%), *Ruminococcaceae* (CON: 7%, HFD: 16%, HPL: 10%, HPH: 12%) and *Lactobacillaceae* (CON: 14%, HFD: 5%, HPL: 12%, HPH: 7%). At the genus level (Figure 7C), the top four dominant genera in all groups were *Blautia* (CON: 2%, HFD: 16%, HPL: 9%, HPH: 12%), *Lactiplantibacillu* (CON: 14%, HFD: 5%, HPL: 12%, HPH: 7%), *Turcibacter* (CON: 2%, HFD: 3%, HPL: 4%, HPH: 5%) and *Faevailbaculum* (CON: 1%, HFD: 3%, HPL: 4%, HPH: 6%). Based on the linear discriminant analysis effect size (LEfSe) and a threshold set at LDA score >4, differences at multiple levels showed that the CON group had significantly enriched taxa belonging to the phylum *Bacteroidetes*, class *Bacteroidia*, order *Bacteroidales*, families *Muribaculaceae* and *Lactobacillaceae*, and genera *Lactiplantibacillus*, *Roseburia* and *ASF356*. The HFD group had significantly enriched taxa belonging to the phylum *Firmicutes*, class *Clostridia*, order *Clostridiales*, families *Peptostreptococcaceae*, *Peptococcaceae* and *Ruminococcaceae*, genera *Romboutsia*, *Clostrdium_senu_stricto_1*, *Ruminiclostridium_9* and *Blautia*, and species *Lachnospiraceae_bacterium_609* (Figure 7D). When the LP1008 treatment groups were compared with the HFD group, the HFD group had significantly enriched taxa belonging to the class *Clostridia*, order *Clostridiales* and family *Lachnospiraceae* (Figure 7E). When the LDA score was >3, the HFD group had more abundant taxa belonging to the *genera Muribaculum*, *Anaerotruncus*, GCA_900066575 and *Ruminococcaceae*_UCG_010. The HPL group had a significantly increased abundance of taxa belonging to the phylum *Proteobacteria*, class *Gammaproteobacteria*, order *Enterobacteriaceae*, family *Enterobacteriaceae*, and genera *Agathobacter* and *Escherichi_Shigella*. The HPH group had a significantly increased abundance of taxa belonging to the genera *Erysipelatoclostridum* and *Proteus* (Figure 7F). Compared with those in the HPL and HPH groups, the dominant bacteria in the HPL group were *Agathobacter* and *Ruminiclostridium*_6, whereas those in the HPH group were *Ruminiclostridium*_9 and *Erysipelatoclostridum* (Figure 7G). Additionally, when the LP1008 treatment groups were compared with the CON group, the dominant bacteria in the CON group were from the phylum *Bacteroidetes*, class *Bacteroidia*, order *Bacteroidales*, family *Muribaculaceae*, and genera *Roseburia* and *Lachnospiraceae_NK4A136_group*. The HPL group had significantly enriched taxa belonging to the phylum *Firmicutes*, families *Peptostreptococcaceae*, *Peptococcaceae* and *Streptococcaceae*, and genera *Romboutsia* and *Clostrdium_senu_stricto_1*. Meanwhile, the dominant bacteria in the HPH group were from the class *Erysipelotrichia*, order *Erysipelotrichales*, family *Erysipelotrichaceae*, and genera *Blautia*, *Faecailbaculum* and *Ruminiclostridium_9* (Figure 7H).

The fecal short-chain fatty acid (SCFA) contents in aged and obese mice were measured, and the concentrations of acetic, propionic and butyric acids were found to be significantly lower in the HFD group than in the CON group. However, there were no significant differences between the LP1008-treated and HFD groups (Figure 8).

### 3.7. Effects of LP1008 on Testicular Redox Status in Aged and Obese Mice

Potential mechanisms underlying impaired reproductive function include oxidative stress, inflammation and apoptosis. HFD-fed mice presented significantly reduced SOD and GPx activity and significantly elevated MDA activity. The 8-week treatment with low and high doses of LP1008 promoted an increase in GPx and a reduction in MDA, whereas there was an obvious increase in CAT only in the HPH group (Figure 9).

### 3.8. Effects of LP1008 on Serum LPS Levels, Testicular Inflammation and Apoptosis-Related Proteins in Aged and Obese Mice

Notably, HFD-fed mice exhibited testicular inflammation and apoptosis, as indicated by increased serum LPS levels; NF-κB, TNF-α, Bax and Bcl-xl expression; and cleaved caspase 3 protein levels (Figure 10 and Figure 11). The expression of inflammatory and apoptotic proteins was significantly downregulated after treatment with low or high doses of LP1008. Furthermore, serum LPS levels were restored after treatment with a high dose of LP1008.

## 4. Discussion

In the present study, obesity in middle-aged mice fed an HFD caused biochemical abnormalities, pathological damage, altered gut microbiota, spatial memory impairment, reproductive dysfunction, increased testicular oxidative stress and apoptosis with inflammation. Meanwhile, treatment with LP1008 suppressed oxidative stress, apoptosis and inflammation; altered the gut microbiome; improved spatial memory; reversed pathologies; and restored reproductive function. Previous studies have shown that aged mice fed a long-term HFD exhibited weight gain and abnormal biochemical parameters [18,19]. Although some *L. plantarum* strains have been shown to exert anti-obesity effects in animal models of obesity [20,21,22], LP1008 had no impact on weight gain, but reduced serum insulin and TC levels, which is similar to the findings of Salaj et al. [23] and could be partially explained by the greater fat absorption ability of some probiotic strains as reported by Yin et al. [24].

Additionally, LP1008 exerted protective effects against liver damage. *L. plantarum* strains have been reported to have a hepatoprotective effect in diet-induced obese mice via the regulation of (1) the gut microbiome, (2) inflammation through the repair of gut barrier-related tight junction proteins by reducing the mRNA expression of proinflammatory mediators, and (3) oxidative stress by increasing the gene expression of antioxidants [25,26]. He et al. have reported that *L. plantarum* JS19-fermented dairy significantly reversed D-galactose-induced premature aging-related liver injury via antioxidant effects, such as increasing the levels of hepatic antioxidants and reducing lipid peroxidation [27].

In the Morris water maze test, spatial learning and memory dependent on the hippocampus were assessed. Reference memory—as a measure of long-term memory—assesses the capacity to retain information for an extended period, whereas working memory—as a measure of short-term memory—assesses the capacity to retain information for a short period [28]. Previous reviews have established an association between obesity and an increased risk of brain decline, manifested through alterations in brain structure (affecting the cortical, hippocampal and cerebellar domains) and function (affecting the cognitive and motor domains). Notably, these reviews also indicated that midlife obesity may have more deleterious effects on cognitive function than late-life obesity [29,30,31]. Both aging and obesity contribute to chronic inflammation, which increases cytokine secretion, damages cerebral structure and function, induces neuroinflammation and leads to cognitive impairment [30]. The present results showed that both long- and short-term memory were disrupted in aged and obese mice, similar to the findings of previous studies [19,32]; furthermore, LP1008 treatment restored both long- and short-term memory, which aligns with previous findings in aged mice [15]. Nie et al. have reported that D-galactose-induced aged mice treated with *L. plantarum* MWFLp-182 showed improved cognitive ability, which may be partly attributable to the modulation of factors involved in inflammation and redox status in the circulation, hippocampus and colon, leading to an improved intestinal barrier and enhanced spatial learning and memory [33]. Gut dysbiosis may be implicated in increased inflammation and neuroinflammation through the gut–brain axis via the release of LPS and bacterial breakdown products [29,34,35]. Additionally, Blair et al. have reported that the activation of androgen and estrogen receptors in the brain by testosterone is necessary for neurological health and cerebral function [36].

Globally, approximately one in ten aging people are predicted to suffer from sarcopenic obesity, which is characterized by the coexistence of low muscle mass and strength and physical inactivity (sarcopenia) with excess fat accumulation [37]. In the present study, LP1008 increased muscle mass and significantly increased the cross-sectional area in aged and obese mice. Lee et al. have reported unchanged muscle weight and increased muscle strength and glycogen levels in aged mice treated with *L. plantarum* TWK10, suggesting that LP1008 plays a role in glycogen synthesis, which affects muscle quality [15]. Additionally, age-related muscular degeneration may be blocked by increasing testosterone levels through the regulation of survival and death pathways, including the Notch, JNK and Akt signaling pathways [38].

As shown in this study, obese mice presented decreased sperm quality, impaired spermatogenesis, and decreased serum testosterone levels and protein expression of enzymes involved in testosterone production (i.e., StAR and 17β-HSD). Obesity- and aging-induced decreases in testosterone levels may be due to (1) senescent Leydig cells or (2) the inhibition of enzymes involved in testosterone biosynthesis. Luo et al. have reported that obesity alters the MAPK pathway, enhances oxidative stress and inflammation, and accelerates Leydig cell aging through reducing the number and functions of cells [39]. Increased reactive oxygen species accumulation in the testes may result in malfunctioning mitochondria, subsequently affecting mitochondrial enzymes—including StAR and CYP11A1—which regulate initial testosterone production [40]. The presented results indicated significantly downregulated 17β-HSD expression, which may trigger decreased testosterone levels due to its role in controlling the last step of testosterone production, which involves the conversion of androstenedione to testosterone [40]. Excess adipose tissue leads to increased aromatase activity and promotes the conversion of testosterone to estrogen. Estrogen further impairs hypothalamus and anterior pituitary function, suppresses the HPG axis and reduces testosterone levels [12]. The obese mice that were administered LP1008 presented improved sperm quality, spermatogenesis and testosterone levels with increased enzyme protein expression (HPL: 17β-HSD; HPH: StAR, CYP17A1, and 17β-HSD). These findings are partially consistent with those reported in other animal models of reproductive dysfunction that have been used to investigate diabetes [41] and exposure to endocrine-disrupting chemicals [42]. These improvements are largely attributed to the modulation of antioxidant imbalance and inflammation.

On the other hand, testosterone is essential for spermatogenesis due to its role in maintaining the blood–testis barrier and adherence between spermatids and Sertoli cells by promoting meiosis and mature sperm release [43]. Additionally, mechanisms such as oxidative stress, inflammation (release of cytokines, such as TNF-α and IL-6), and apoptosis induced by aging and obesity impair testosterone synthesis, spermatogenesis, and sperm quality [44,45]. The presented results demonstrated that LP1008 treatment restored reproductive function by increasing testicular antioxidant levels, reducing lipid peroxidation and apoptotic and inflammatory cytokine protein expression, and increasing testosterone synthesis-related enzymes.

The gut microbiota has become a novel therapeutic agent for various diseases, and gut dysbiosis is associated with obesity and aging [15,25]. In the present study, aged and obese mice had a significantly increased *F/B* ratio, which is consistent with findings in other mouse models of obesity [21] and aging [27]. Modest modulation of the gut microbiota was observed with LP1008 treatment, which is partially consistent with the findings of previous studies in which either the same or similar strains were used in aged and obese animal models [15,20,21,22,27]. LP1008 treatment reversed the obesity- and aging-induced reduction in the *F*/*B* ratio, possibly due to a decrease in bacteria belonging to the *Firmicutes* phylum. The *F*/*B* ratio is considered a hallmark of obesity, and an elevated *F*/*B* ratio has been associated with upregulated appetite in obese patients, which may be due to enhanced energy harvesting and low-grade inflammation [46,47]. The HFD group also presented relatively lower abundances of *Muribaculaceae* and *Roseburia* and increased abundances *of Blautia*, *Peptococcaceae*, *Ruminococcaceae, Peptostreptococcaceae, Romboutsia* and *Clostridium sensu stricto 1*. *Muribaculaceae* are recognized as beneficial colonic bacteria, due to their role in SCFA production, maintenance of intestinal barrier function and immune system regulation. The presence of these bacteria has been reported to be inversely correlated with obesity-related indicators in human and animal studies [48]. *Roseburia* is associated with fiber and polysaccharide metabolism and SCFA production [49]; however, its associations with obesity and aging remain controversial [50]. *Blautia*, *Romboutsia* (*Peptostreptococcaceae*) and *Clostridium sensu stricto 1* are associated with intestinal inflammation and obesity. Zeng et al. have reported that obese patients presented increased abundances of *Blautia* and *Romboutsia*, which were positively correlated with obesity-related indicators [51]. However, aged and obese mice treated with a low dose of LP1008 presented increased *Proteobacteria*—a phylum consistently associated with obesity [52,53]. Juárez-Fernández et al. have reported that elderly individuals had increased *Proteobacteria*, which was positively correlated with inflammatory markers via an increase in intestinal permeability and the stimulation of inflammation [54]. A greater abundance of *Erysipelatoclostridium* was observed in the HPH group. This bacterium, along with its metabolite ptilosteroid A, has been reported as a biomarker of intestinal injury [55]. An increase in *Faecalibacterium* abundance was also observed. A systematic review by Xu et al. on the gut microbiota in obesity and metabolic disorders provided evidence supporting the observation that *Faecalibacterium* is a lean-associated bacterium [52].

Acetic, propionic and butyric acids are iconic SCFAs, and recent findings have shown that an HFD results in notably lower production of these three SCFAs. The decreases in acetic and propionic acid production may be due to a decrease in the abundance of *Bacteroidetes*, which are inhibited by aging and diet-induced obesity [56]. Lower levels of butyric acid were also observed in this study, which is consistent with findings in other HFD-fed animal models [21,22]. SCFAs may improve body weight through glucose and lipid metabolism. Acetate stimulates glucagon-like peptide 1 and peptide YY, lowers appetite, and reduces fatty acid and triglyceride synthesis, thereby reducing adiposity. Propionate decreases gluconeogenesis in the liver, while butyrate increases the concentration of leptin, which controls appetite. In other words, lower SCFA concentrations may result in increased body weight and biochemical abnormalities, as observed in the present study.

The present findings revealed that an HFD impaired ileal morphology and reduced SCFA production. SCFAs—particularly butyrate—can upregulate proteins that modulate tight junctions and intensify the mucus layer covering the epithelial layer, thus maintaining the integrity of the gut barrier. SCFAs may also play beneficial roles in reproductive and cognitive functions [57,58]. However, LP1008 treatment did not lead to any significant increase in SCFA levels, despite improvements in ileal morphology. A previous study using the same strain in aging mice also revealed no significant alterations in SCFA-producing bacteria [15]. Future studies with prolonged treatment durations or higher dosages of LP1008 are warranted, in order to fully explore this phenomenon.

In the present study, LOH not only affected reproductive function but also triggered aging-related pathological deficits, such as cognitive and muscle dysfunction. After treatment with the probiotic LP1008, diet-induced obesity, age-related reproductive dysfunction and pathological damage were counteracted, together with increased testosterone levels, alterations in the gut microbiome and the regulation of mediators involved in oxidative stress, apoptosis and inflammation.

## Figures and Tables

**Figure 1 antioxidants-13-01498-f001:**
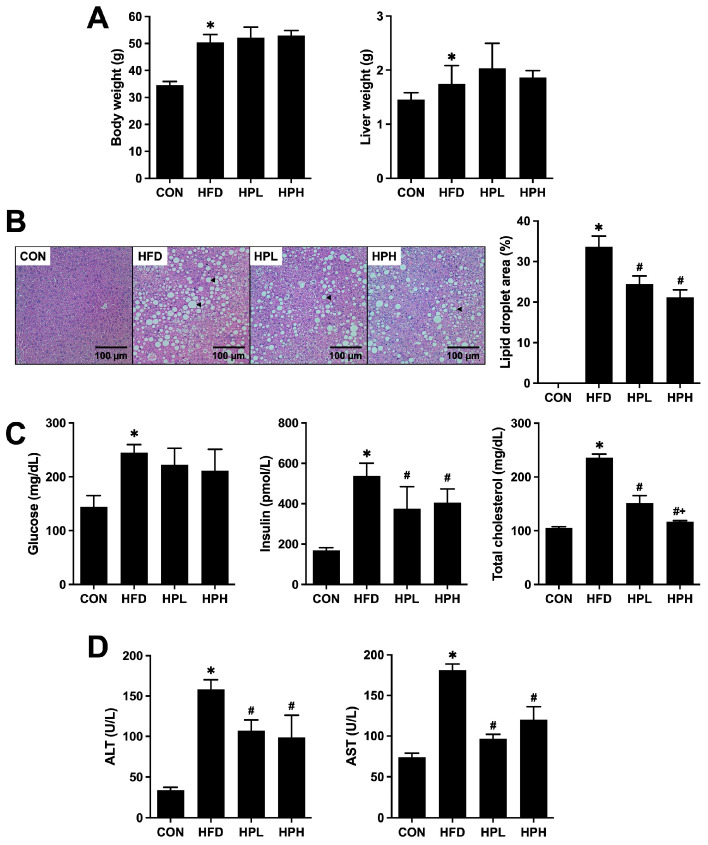
(**A**) Body and liver weights; (**B**) histopathological observations of the liver (arrows indicate hepatic steatosis) and percentages of hepatic lipid droplet areas; and (**C**) serum glucose, insulin, total cholesterol, (**D**) ALT and AST levels in aged and obese mice (*n* = 8 per group). All data are represented as mean  ±  SD. * *p* < 0.05 vs. the CON group; ^#^
*p* < 0.05 vs. the HFD group; ^+^
*p* < 0.05 vs. the HPL group. CON, control; HFD, high-fat diet; HPL, low-dose LP1008, HPH, high-dose LP1008; AST, aspartate transaminase; ALT, alanine transaminase.

**Figure 2 antioxidants-13-01498-f002:**
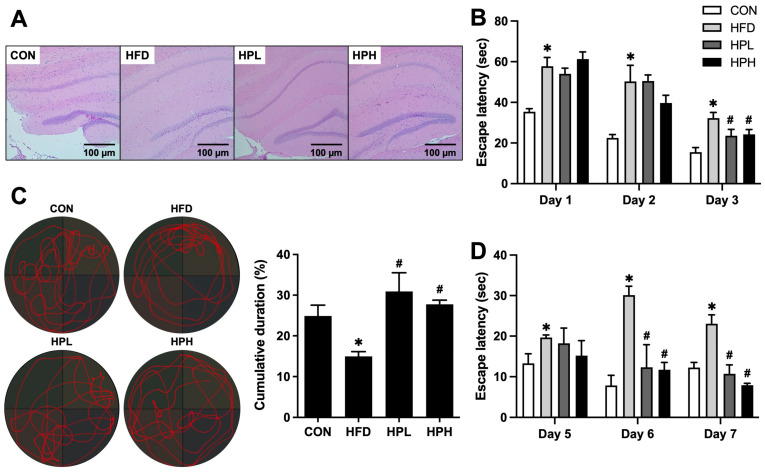
(**A**) Histopathological observations of the brain and spatial learning and memory, including (**B**) the mean escape latency in spatial acquisition training, (**C**) swimming pathways and percentage of time spent in the target quadrant, and (**D**) the mean escape latency in the working memory test in aged and obese mice (*n* = 8 per group). All data are represented as mean  ±  SD. * *p* < 0.05 vs. the CON group, ^#^
*p* < 0.05 vs. the HFD group. CON, control; HFD, high-fat diet; HPL, low-dose LP1008; HPH, high-dose LP1008.

**Figure 3 antioxidants-13-01498-f003:**
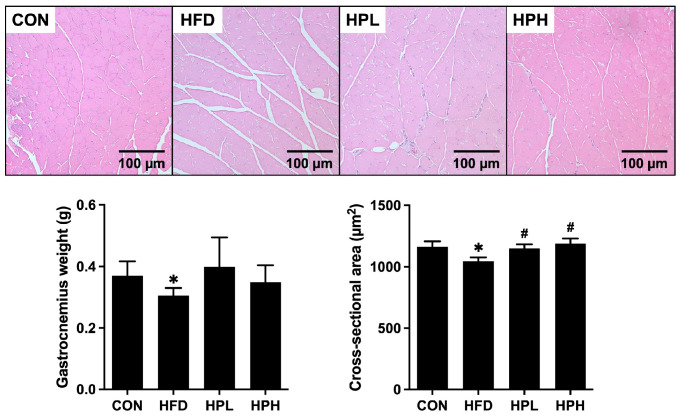
Histopathological observations of muscle and gastrocnemius muscle weights and mean cross-sectional areas in aged and obese mice (*n* = 8 per group). All data are represented as mean  ±  SD. * *p* < 0.05 vs. the CON group, ^#^
*p* < 0.05 vs. the HFD group. CON, control; HFD, high-fat diet; HPL, low-dose LP1008; HPH, high-dose LP1008.

**Figure 4 antioxidants-13-01498-f004:**
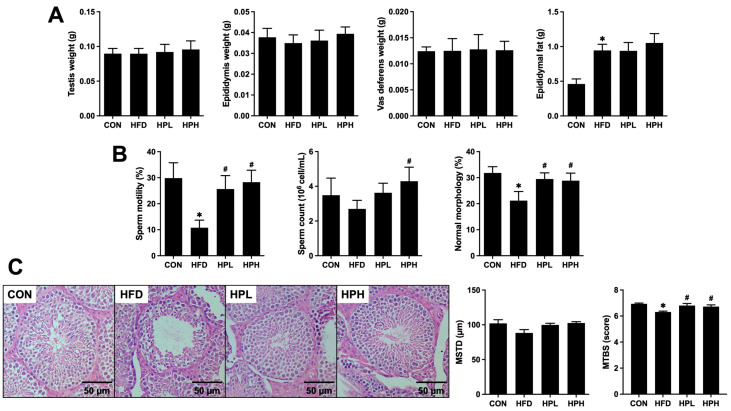
(**A**) Testis, epididymis and vas deferens weights; (**B**) sperm quality, including sperm motility, sperm count and normal morphology rate; and (**C**) histopathological observations of testes with related parameters, including MSTD and MTBS, in aged and obese mice (*n* = 8 per group). All data are represented as mean  ±  SD. * *p* < 0.05 vs. the CON group; ^#^
*p* < 0.05 vs. the HFD group. CON, control; HFD, high-fat diet; HPL, low-dose LP1008; HPH, high-dose LP1008; MSTD, mean seminiferous tubule diameter; MTBS, mean testicular biopsy score.

**Figure 5 antioxidants-13-01498-f005:**
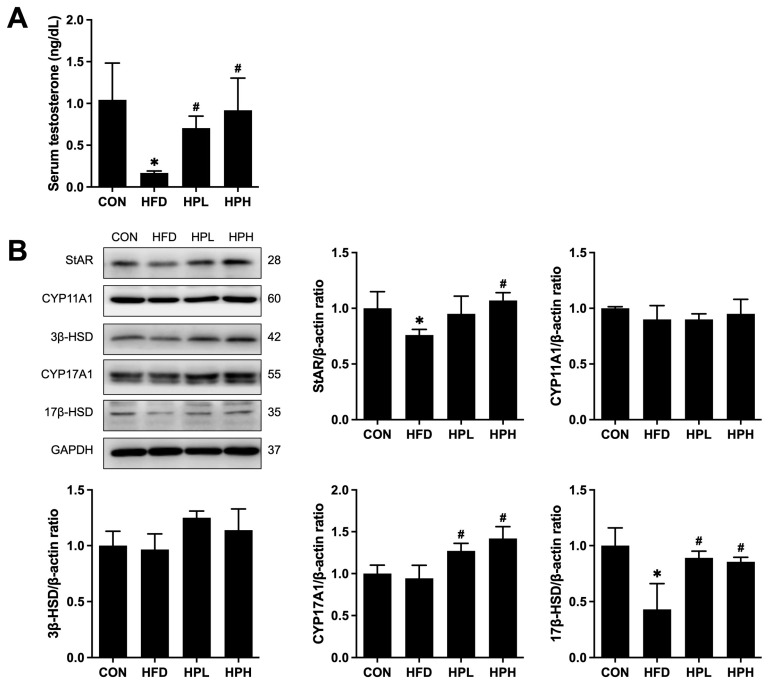
(**A**) Serum testosterone levels; and (**B**) Western blotting and quantitative analysis of the relative protein expression of testicular StAR, CYP11A1, 3β-HSD, CYP17A1 and 17β-HSD in aged and obese mice (**A**: *n* = 8 in per group; **B**: *n* = 4 per group). All data are represented as mean  ±  SD. * *p* < 0.05 vs. the CON group, ^#^
*p* < 0.05 vs. the HFD group. CON, control; HFD, high-fat diet; HPL, low-dose LP1008; HPH, high-dose LP1008. The whole Western blot is shown in Appendix A.

**Figure 6 antioxidants-13-01498-f006:**
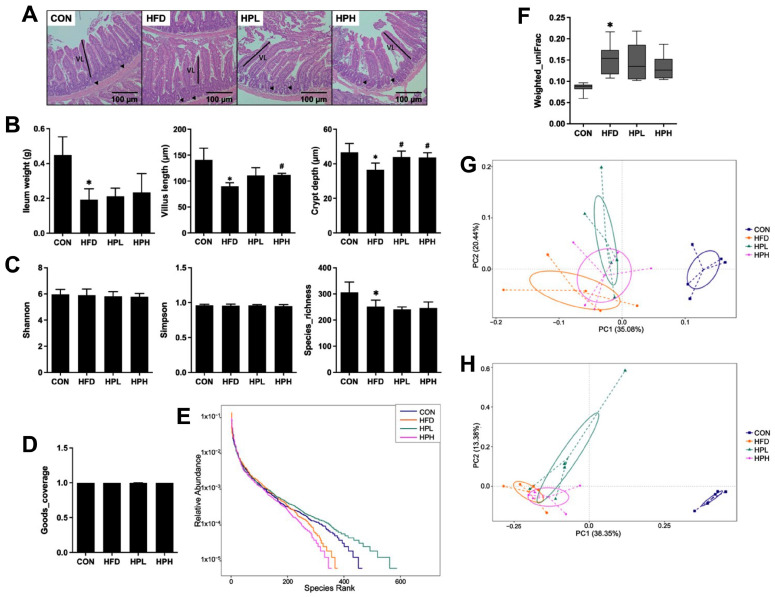
(**A**) Histopathological observations of the ileum; (**B**) ileum weight and related parameters, including villus length and crypt depth (arrows indicated); (**C**) the characteristics of microbial communities, including Shannon, Simpson and species richness; (**D**) Good’s coverage; (**E**) rank abundance curve; (**F**) weighted UniFrac; (**G**) weighted UniFrac-based PCoA; and (**H**) Bray–Curtis dissimilarity-based PCoA in aged and obese mice (**A**,**B**: *n* = 8 per group; **C**–**H**: *n* = 4 per group). All data are represented as mean  ±  SD. * *p* < 0.05 vs. the CON group, and ^#^
*p* < 0.05 vs. the HFD group. CON, control; HFD, high-fat diet; HPL, low-dose LP1008; HPH, high-dose LP1008.

**Figure 7 antioxidants-13-01498-f007:**
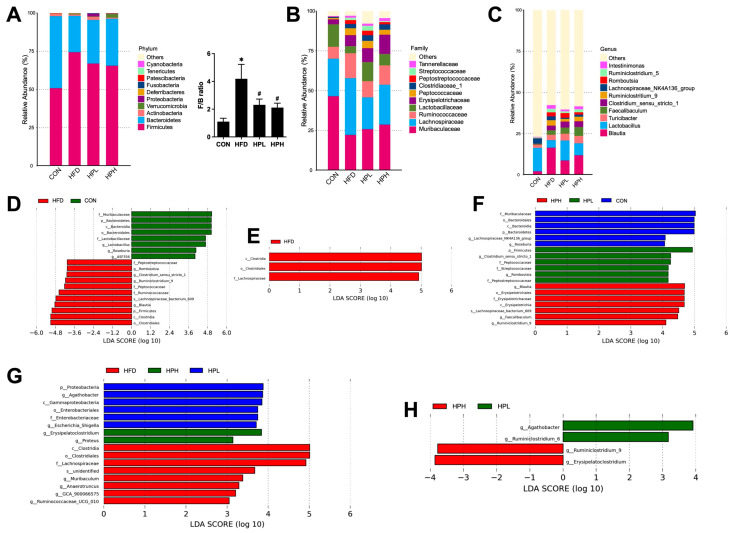
Relative abundances of the microbial community at (**A**) the phylum level with the F/B ratio, (**B**) the family level and (**C**) the genus level. Dominant taxa based on LEFSe at LDA scores >4 in comparisons of (**D**) the CON and HFD groups; (**E**) the HFD, HPL and HPH groups; and (**F**) the CON, HPL and HPH groups. LDA score > 3 in comparisons of the (**G**) HFD, HPL and HPH groups and (**H**) the HPL and HPH groups (*n* = 4 per group). * *p* < 0.05 vs. the CON group and ^#^
*p* < 0.05 vs. the HFD group. CON, control; HFD, high-fat diet; HPL, low-dose LP1008; HPH, high-dose LP1008; F/B, *Firmicutes/Bacteroidetes*.

**Figure 8 antioxidants-13-01498-f008:**
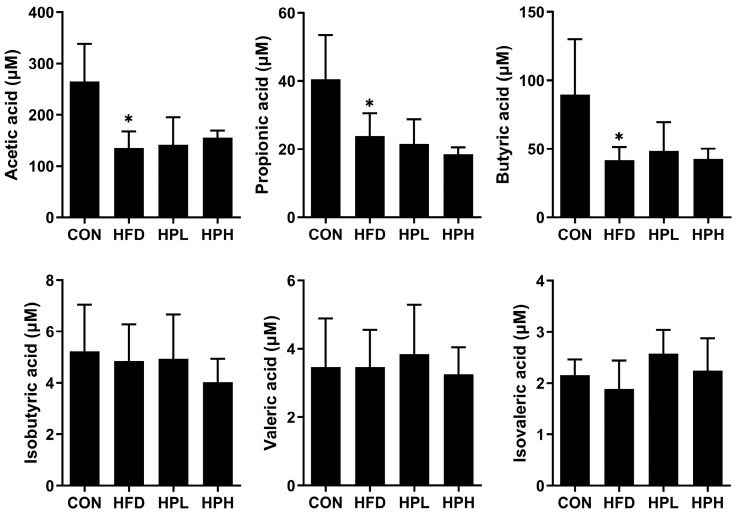
Fecal SCFA contents in aged and obese mice (*n* = 4 per group). All data are represented as mean  ±  SD. * *p* < 0.05 vs. the CON group. CON, control; HFD, high-fat diet; HPL, low-dose LP1008; HPH, high-dose LP1008.

**Figure 9 antioxidants-13-01498-f009:**
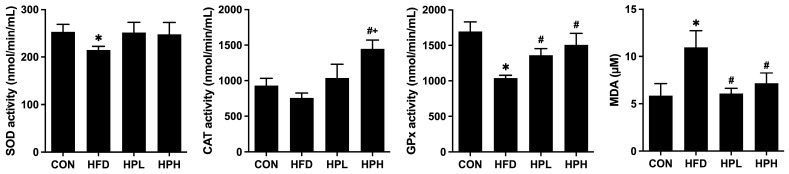
Testicular SOD, CAT and GPx activities and MDA levels in aged and obese mice (*n* = 4 per group). All data are represented as mean  ±  SD. * *p* < 0.05 vs. the CON group, and ^#^
*p* < 0.05 vs. the HFD group, and ^+^
*p* < 0.05 vs. the HPL group. CON, control; HFD, high-fat diet; HPL, low-dose LP1008; HPH, high-dose LP1008; SOD, superoxide dismutase; CAT, catalase; GPx, glutathione peroxidase; MDA, malondialdehyde.

**Figure 10 antioxidants-13-01498-f010:**
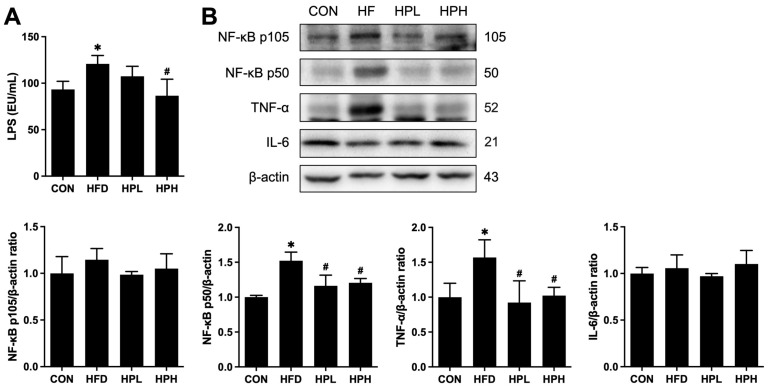
(**A**) Serum LPS levels and (**B**) Western blotting and quantitative analysis of the relative protein expression of testicular NF-κB p105, NF-κB p50, TNF-α and IL-6 in aged and obese mice (**A**: *n* = 8 per group; **B**: *n* = 4 per group). All data are represented as mean  ±  SD. * *p* < 0.05 vs. the CON group, ^#^
*p* < 0.05 vs. the HFD group. CON, control; HFD, high-fat diet; HPL, low-dose LP1008; HPH, high-dose LP1008; LPS, lipopolysaccharide. The whole Western blot is shown shown in Appendix A.

**Figure 11 antioxidants-13-01498-f011:**
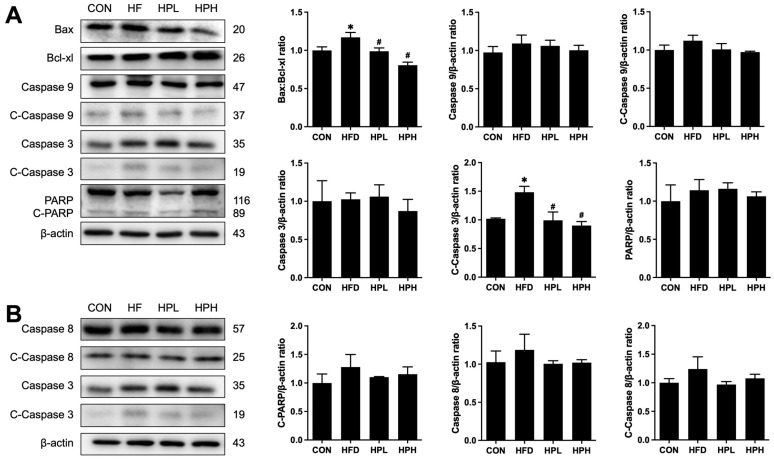
(**A**,**B**) Western blotting and quantitative analysis of the relative protein expression levels of the testicular Bax:Bcl-xl ratio, Caspase 9, C-Caspase 9, Caspase 3, C-Caspase 3, PARP, C-PARP, Caspase 8 and C-Caspase 8 in aged and obese mice (*n* = 4 per group). All data are represented as mean  ±  SD. * *p* < 0.05 vs. the CON group, ^#^
*p* < 0.05 vs. the HFD group. CON, control; HFD, high-fat diet; HPL, low-dose LP1008; HPH, high-dose LP1008. The whole Western blot is shown in Appendix A.

**Table 1 antioxidants-13-01498-t001:** Composition of the experimental diets.

Nutrients	5001	D12451
Carbohydrate (%)	48.1	40.3
Protein (%)	24.1	23.7
Fat (%)	5.1	23.4
Fiber (%)	5.3	5.8
Energy from carbohydrate (% of kcal)	57	35
Energy from protein (% of kcal)	14	20
Energy from fat (% of kcal)	29	45
Metabolizing energy (kcal/g)	2.86	4.73

## Data Availability

All data generated or analyzed during this study are included in this article.

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
