# Peer review of "Lactiplantibacillus plantarum 1008 Promotes Reproductive Function and Cognitive Activity in Aged Male Mice with High-Fat-Diet-Induced Obesity by Altering Metabolic Parameters and Alleviating Testicular Oxidative Damage, Inflammation and Apoptosis"

_antioxidants, 2024, doi:10.3390/antiox13121498_

Round 1
Reviewer 1 Report
A nice article, but needs some modifications.
Reviewer’s Comments
The article by Liu et. al. investigates the effects of Lactiplantibacillus plantarum 1008 (LP1008) on reproductive function and cognitive activity in aged male mice with high-fat diet (HDF)-induced obesity. This is a fantastic follow up study in recognition of previous studies noting LP1008’s benefits on cognitive decline and muscle atrophy due to aging, but this study specifically explores its impact on age and obesity-related late-onset hypogonadism (LOH) , a condition marked by reduced testosterone and reproductive function. The authors analyzed mouse key parameters like: metabolic and liver health, cognitive function, muscle and reproductive health, oxidative stress, inflammation, gut microbiota etc. HFD causes abnormalities in biochemical, pathological, altered microbiota, spatial memory impairment, reproductive dysfunction, increased testicular oxidative stress, and apoptosis with inflammation. Treatment with LP1008 suppressed oxidative stress, apoptosis, and inflammation; altered the gut microbiome; improved spatial memory; reversed pathologies; and restored reproductive function. Their findings suggest that LP1008 could offer a promising approach to mitigate age and obesity-related health issues. This article is well written and articulated but the following points should be considered to improve the quality and acceptance to general readers.
The following points should consider:
1. In Fig.1B, missing scale bar in image (in other Figs. as well), graphical presentation of decreased fat accumulation may be done by counting fat droplets.
2. In Fig 6A, colonic crypts should be presented from the colonic section, not by cecal or intestinal sections. What is shown in the image is only villi. Crypt should be presented by clear indication, even bowing up, indicating by arrow. Explain how crypt depth was measured? (in line 284-).
3. The author should increase consistent font size in all figs; fonts in figs. 2 and 8 are acceptable, others should be increased in bold phase.
Finally, discussion may be improved by taking a global perspective; not merely in a single country.
Author Response
Dear Professor Editor:
Please find enclosed our revised original paper entitled “Lactiplantibacillus plantarum 1008 promotes reproductive function and cognitive activity in aged male mice with high-fat diet-induced obesity by altering metabolic parameters and alleviating testicular oxidative damage, inflammation and apoptosis”. We appreciated the reviewers’ comments and suggestions provided to further improve our manuscript.
Sincerely yours
Chih-Wei Tsao, MD., Ph.D.
Division of Urology, Department of Surgery, Tri-Service General Hospital, National Defense Medical Center
No. 325, Section 2, Cheng-Gung Road,
Neihu, Taipei 114, Taiwan
Telephone: +886-2-87927170
Fax: +886-2-87927172
e-mail: weisurger@gmail.com
Reviewer 1 Comments
The article by Liu et. al. investigates the effects of Lactiplantibacillus plantarum 1008 (LP1008) on reproductive function and cognitive activity in aged male mice with high-fat diet (HDF)-induced obesity. This is a fantastic follow up study in recognition of previous studies noting LP1008’s benefits on cognitive decline and muscle atrophy due to aging, but this study specifically explores its impact on age and obesity-related late-onset hypogonadism (LOH) , a condition marked by reduced testosterone and reproductive function. The authors analyzed mouse key parameters like: metabolic and liver health, cognitive function, muscle and reproductive health, oxidative stress, inflammation, gut microbiota etc. HFD causes abnormalities in biochemical, pathological, altered microbiota, spatial memory impairment, reproductive dysfunction, increased testicular oxidative stress, and apoptosis with inflammation. Treatment with LP1008 suppressed oxidative stress, apoptosis, and inflammation; altered the gut microbiome; improved spatial memory; reversed pathologies; and restored reproductive function. Their findings suggest that LP1008 could offer a promising approach to mitigate age and obesity-related health issues. This article is well written and articulated but the following points should be considered to improve the quality and acceptance to general readers.
The following points should consider:
- In Fig.1B, missing scale bar in image (in other Figs. as well), graphical presentation of decreased fat accumulation may be done by counting fat droplets.
Our response:
Thank you for your professional comment. We have added scale bar in all figures, and incorporated your comment of fat droplets assessment with related description in figure 1B and section 3.1.
- In Fig 6A, colonic crypts should be presented from the colonic section, not by cecal or intestinal sections. What is shown in the image is only villi. Crypt should be presented by clear indication, even bowing up, indicating by arrow. Explain how crypt depth was measured? (in line 284-).
Our response:
Thank for your important opinion. We reconfirmed the tissue sections used in histological assessment and gut microbiota. The villus lengths and crypt depths were quantified using hematoxylin and eosin-stained distal ileal sections by Image J. We have rewritten the related sections in the article and added arrows in representative images of each group to indicate the colonic crypts (Figure 6A).
- The author should increase consistent font size in all figs; fonts in figs. 2 and 8 are acceptable, others should be increased in bold phase.
Our response:
Thank you for your helpful suggestion. We have revised font size in all figures.
Finally, discussion may be improved by taking a global perspective; not merely in a single country.
Our response:
Thank you for your academic comment. According to the comments of reviewer 1 and reviewer 3, we finally omitted the description of Taiwan epidemiological statistics & rewrote the paragraph in “Discussion” part.

Reviewer 2 Report
This manuscript shows very interesting results on the effect of a probiotic strain on obesity induced health issues in a mouse model. Results are presented clearly in Figures, the text is well written for part of the manuscript but especially the results section needs improvement. The description of experimental data is often unclear or misleading, e.g. the use of the word however (lines 190-193) when there is no contrast. Also the usage of both percentages and fold changes in the same sentence and concerning the same data is confusing and difficult to understand/interpret as a reader. The results section therefore should be rewritten to clarify these aspects
Lines 190-193: however, but no contrast
line 201: however: in this case there is a contrast but this is what the authors probably are looking for (efficacy of treatment) so I would advice to focus on this
Line 217: additionally: this is not additionally but describes data which cannot be directly compared
Lines 269-271: percentage change and fold change in one sentence
lines 273-275: the same, also with does a 1.0 fold change mean?
Author Response
Dear Professor Editor:
Please find enclosed our revised original paper entitled “Lactiplantibacillus plantarum 1008 promotes reproductive function and cognitive activity in aged male mice with high-fat diet-induced obesity by altering metabolic parameters and alleviating testicular oxidative damage, inflammation and apoptosis”. We appreciated the reviewers’ comments and suggestions provided to further improve our manuscript.
Sincerely yours
Chih-Wei Tsao, MD., Ph.D.
Division of Urology, Department of Surgery, Tri-Service General Hospital, National Defense Medical Center
No. 325, Section 2, Cheng-Gung Road,
Neihu, Taipei 114, Taiwan
Telephone: +886-2-87927170
Fax: +886-2-87927172
e-mail: weisurger@gmail.com
Reviewer 2 Comments
Major comments
This manuscript shows very interesting results on the effect of a probiotic strain on obesity induced health issues in a mouse model. Results are presented clearly in Figures, the text is well written for part of the manuscript but especially the results section needs improvement. The description of experimental data is often unclear or misleading, e.g. the use of the word however (lines 190-193) when there is no contrast. Also the usage of both percentages and fold changes in the same sentence and concerning the same data is confusing and difficult to understand/interpret as a reader. The results section therefore should be rewritten to clarify these aspects.
Detail comments
Lines 190-193: however, but no contrast
Our response:
Thank you for your professional comment. We expected that probiotic treatment might have a weight-lowering effect, but the mean body weight in the HPL and HPH groups after 8-week treatment were similar to the HFD group (non-probiotic treatment). We have rewritten this paragraph to be more in line with your opinion (section 3.1.).
line 201: however: in this case there is a contrast but this is what the authors probably are looking for (efficacy of treatment) so I would advice to focus on this
Our response:
Thank you for your helpful suggestion. We have revised it in section 3.1.
Line 217: additionally: this is not additionally but describes data which cannot be directly compared
Our response:
Thank you for your kindly reminder. We have considered this comment and revised the paragraph in section 3.2.
Lines 269-271: percentage change and fold change in one sentence
Our response:
Thank you for your professional suggestion. We have revised the paragraph in section 3.5.
lines 273-275: the same, also with does a 1.0 fold change mean?
Our response:
Thank you for your kindly reminder. We have revised the paragraph and corrected the mistake (1.0 fold but actually 2.0 fold) in section 3.5.

Reviewer 3 Report
Thank you for opportunity to review manuscript antioxidants-3283766 entitled: ‘Lactiplantibacillus plantarum promotes reproductive function and cognitive activity in aged male mice with high-fat diet-induced obesity’.
Paper focuses on effects of high-fat diet induced metabolic disturbances in old male mice and if those disturbances can be counteracted with administration of Lactiplantibacillus plantarum 1008 for 8 weeks period.
General: Please, check that abbreviation is consistent throughout the text. Include animal number per each group in every figure legend (e.g. CON, n=XX etc)
I got an impression that you compared CON to HDF and then every group to HFD, please, explain better in the M&M which groups you compared. Include also in the legends how the data is presented, e.g. Mean ± SD.
Please, keep the same description of groups, e.g. in figures 5B, group HFD displayed as HF. Please, correct.
I also highly recommend to check the English. For instance, in line 485 … you wrote: may trigger appetite… However, ratio F/B itself cannot do any trigger. It would be more correct to say: increased F/B ratio has been shown to have association with upregulated appetite in obese patients… etc.
I also recommend to write more clear conclusion. Lines 539-531 belong to introduction in my opinion. Line: 534 … pathological damage were reversed because of increased testosterone levels … I would recommend to change to … pathological damage were counteracted together with increased testosterone levels that …
Specific comments:
Title: Lines 1-3
I recommend to introduce the strain number 1008 already in the title to emphasis usage of novel L.p strain.
I also recommend to rephrase the title taking in account metabolic disturbances and inflammatory changes and antioxidative capacity, not only reproductive function and cognitive activity.
Abstract:
Indicate age of animals and number of mice per each treatment group. Lines 16-17
Please, use ‘gastrocnemius muscle’ instead of ‘gastrocnemius’ Line 20
Keywords: Lactiplantibacillus plantarum is already mentioned in the title.
Introduction: sufficient
Materials and Methods:
2.1. Animals and treatment:
Please, include information about acclimatization time on line 76. How many weeks?
Lines 85-88.
How many animals were in control diet, HFD, HPL group, HPH group? Include this information. Have any mice naturally died during duration of the study or were killed by cage-mates? I suggest to include a schematic illustration for experimental flow as a supplement figure.
How did animals receive bacterial supplementation for 8-weeks duration period?
How was blood collected and into which tubes? Which speed and temp. was used for centrifugation?
Which organs have you dissected to measure organ wt? For instance you do not specify dissection of epididymal fat mass, but you mention this in result section.
Lines 119-120. please, include reference paper for protocol
Did you perform centrifugation of lysates? Line 140
Line 114: How were tissues’ biopsies stained direct after fixation? Which technique have you use for tissue sectioning? What about the thickness of the tissue samples? Which method you used for mounting of the coverslip? Describe missing information.
Lines 118-135. Which system was used for videorecording and data analysis? Specify in the text.
Lines 161-167. Fecal microbial profiling: Why did you not collect cecal content at the dissection? How were fresh faeces collected to avoid cross contamination from cage-mates? How were the samples frozen? On dry ice or liquid nitrogen? How DNA was extracted? which kit and which instrument was used for DNA sequencing? how the DNA library was prepared? Etc. Please, specify this in the text.
Please, describe which bioinformatic tools were applied for sequenced data analysis and which database was used for the OTU identification?
The raw sequencing data or OTU table need to be provided as e.g. PubMed project.
Line 169 Frozen faecal samples were extracted… Please, describe the method or include reference.
Line 177 -184.
Have you used the same tools for microflora composition and function?
3. Results:
Figure 1A. Line 204.
You describe liver damage with lipid accumulation; however you have not performed lipid staining such as Oil Red O.
Please, include the reference for H&E method that indicate the same. Show with arrows on the image fat accumulation. Indicate the scale bar for each image you included.
Figure 2. (A), Line 229.
I have doubts about depth of the brain tissue biopsy you show for each group. Please, indicate which mice brain matrice you used to dissect hippocampal area? Which cerebral hemisphere? Indicate the scale bar for each image you included. It looks like HFD image was taken with different magnification.
Lines 234-243. Please, specify in M&M how did you dissect out gastrocnemius muscle. From which side, left or right, gastrocnemius muscle was dissected and measured?
Figure 4. line 259. Indicate the scale bar for each image you included
Lines 281-301. The images authors show represent histological features of the small intestine (clear crypt-villus unit) but not cecum. Since authors have not specified the exact place of the biopsy collection, I assume that results might represent different parts from different groups and therefore cannot be compared.
Cecum is the part of large intestine and does not have villi. Please, check references for histological features of small and large intestines in mice.
https://currentprotocols.onlinelibrary.wiley.com/doi/full/10.1002/cpz1.1062
https://journals.sagepub.com/doi/full/10.1258/la.2009.009112
Figure 6 B. Are you presenting cecum with or without content? Please, specify in the text.
Discussion:
Line 394: … middle-aged mice...
Should mice of 74 weeks of age be considered as old mice? Do you have reference?

Author Response
Dear Professor Editor:
Please find enclosed our revised original paper entitled “Lactiplantibacillus plantarum 1008 promotes reproductive function and cognitive activity in aged male mice with high-fat diet-induced obesity by altering metabolic parameters and alleviating testicular oxidative damage, inflammation and apoptosis”. We appreciated the reviewers’ comments and suggestions provided to further improve our manuscript.
Sincerely yours
Chih-Wei Tsao, MD., Ph.D.
Division of Urology, Department of Surgery, Tri-Service General Hospital, National Defense Medical Center
No. 325, Section 2, Cheng-Gung Road,
Neihu, Taipei 114, Taiwan
Telephone: +886-2-87927170
Fax: +886-2-87927172
e-mail: weisurger@gmail.com
Reviewer 3 Comments
Major comments
General: Please, check that abbreviation is consistent throughout the text. Include animal number per each group in every figure legend (e.g. CON, n=XX etc)
Our response:
Thank you for your kindly reminder. We have rechecked the abbreviation and added animal numbers per group in each figure legend.
I got an impression that you compared CON to HDF and then every group to HFD, please, explain better in the M&M which groups you compared.
Our response:
Thank you for your professional comment. We have supplemented the statistical analysis with explanations of comparison groups in section 2.10.
Include also in the legends how the data is presented, e.g. Mean ± SD.
Our response:
Thank you for your academic opinion. We have added the descriptive statistics for numerical variables in each figure legend.
Please, keep the same description of groups, e.g. in figures 5B, group HFD displayed as HF. Please, correct.
Our response:
Thank you for your kindly reminder. We have rechecked the abbreviation in all figures.
I also highly recommend to check the English. For instance, in line 485 … you wrote: may trigger appetite… However, ratio F/B itself cannot do any trigger. It would be more correct to say: increased F/B ratio has been shown to have association with upregulated appetite in obese patients… etc.
Our response:
Thank you for your professional comment. We have revised the statement as “The F/B ratio is considered a hallmark of obesity and an elevated F/B ratio has been associated with upregulated appetite in obese patients, which may be due to enhanced energy harvesting and low-grade inflammation.”
I also recommend to write more clear conclusion. Lines 539-531 belong to introduction in my opinion.
Our response:
Thank you for your academic opinion. According to the comments of reviewer 1 and reviewer 3, we finally omitted the description of Taiwan epidemiological statistics & rewrote the paragraph in “Discussion” part.
Line: 534 … pathological damage were reversed because of increased testosterone levels … I would recommend to change to … pathological damage were counteracted together with increased testosterone levels that …
Our response:
Thank for your professional comment. We agree with your perspective and revised the statement as “After treatment with the probiotic LP1008, diet-induced obesity, age-related reproductive dysfunction and pathological damage were counteracted, together with increased testosterone levels, alterations in the gut microbiome and the regulation of mediators involved in oxidative stress, apoptosis and inflammation.”
Detail comments
Specific comments:
Title: Lines 1-3
I recommend to introduce the strain number 1008 already in the title to emphasis usage of novel L.p strain.
Our response:
Thank you for your professional comment. We have added the strain number into the title.
I also recommend to rephrase the title taking in account metabolic disturbances and inflammatory changes and antioxidative capacity, not only reproductive function and cognitive activity.
Our response:
Thank you for your professional opinion. We have incorporated this comment into the title.
Abstract:
Indicate age of animals and number of mice per each treatment group. Lines 16-17
Please, use ‘gastrocnemius muscle’ instead of ‘gastrocnemius’ Line 20
Keywords: Lactiplantibacillus plantarum is already mentioned in the title.
Our response:
Thank you for your helpful suggestion. We have revised the abstract.
Introduction: sufficient
Materials and Methods:
2.1. Animals and treatment:
Please, include information about acclimatization time on line 76. How many weeks?
Our response:
Thank you for your professional comment. We have incorporated this comment in section 2.1. The mice were divided into control diet or high-fat diet after 1-week adaptation.
Lines 85-88.
How many animals were in control diet, HFD, HPL group, HPH group? Include this information. Have any mice naturally died during duration of the study or were killed by cage-mates? I suggest to include a schematic illustration for experimental flow as a supplement figure.
Our response:
Thank you for providing these insights. We have incorporated animal numbers. There were 8 mice in each group, and none of them died during the study. We found wounds in two mice in the control diet group. They partially recovered after being housed individually, and both survived until euthanasia. The schematic experimental flow has been provided as a supplementary figure.
How did animals receive bacterial supplementation for 8-weeks duration period?
Our response:
The 8-week supplementation was administered by oral gavage daily, and we have added this information.
How was blood collected and into which tubes? Which speed and temp. was used for centrifugation?
Our response:
The blood was collected carefully from the left ventricle with a 26-gauge needle attached to a 1 mL syringe, and aspirated into the 1.5 mL Eppendorf tube slowly to prevent hemolysis. The collected blood was centrifuged for 10 min at 3000 x g and 4°C to separate serum after standing for 20 minutes to clot. We have incorporated this information in the section 2.2.
Which organs have you dissected to measure organ wt? For instance you do not specify dissection of epididymal fat mass, but you mention this in result section.
Our response:
Thank you for your academic comment. We measured weights of liver, kidney, ileum, gastrocnemius muscle, testis, epididymis, vas deferens and epididymal fat mass, and we have reflected this comment in the section 2.1.
Lines 119-120. please, include reference paper for protocol
Our response:
Thank you for providing this comment. We have added reference paper (Reference 17) in section 2.5.
Did you perform centrifugation of lysates? Line 140
Our response:
The lysates were centrifuged at 4°C and 14000 rpm for 15 min, and we have added this information in section 2.6.
Line 114: How were tissues’ biopsies stained direct after fixation? Which technique have you use for tissue sectioning? What about the thickness of the tissue samples? Which method you used for mounting of the coverslip? Describe missing information.
Our response:
Thank you for providing this comment. We have added missing information in the section 2.4.
Lines 118-135. Which system was used for videorecording and data analysis? Specify in the text.
Our response:
Thank you for your suggestion. The tracking system has added in section 2.5.
Lines 161-167. Fecal microbial profiling: Why did you not collect cecal content at the dissection? How were fresh faeces collected to avoid cross contamination from cage-mates? How were the samples frozen? On dry ice or liquid nitrogen? How DNA was extracted? which kit and which instrument was used for DNA sequencing? how the DNA library was prepared? Etc. Please, specify this in the text.
Our response:
Thank you for providing this comment. Fecal and cecal microbiota were common tools for examining the gut microbiome in HFD-induced rodent model. Lee et al. conducted a high-fat diet-induced obesity model in B6 mice and compared fecal and cecal microbiota. It is concluded that rather than sample source (feces or cecum), dietary pattern showed a more significant impact on changes in gut microbiome (REF1). Yan et al. established a high-fat diet-induced NAFLD model in B6 mice, and reported that the variation of gut microbiota in the cecum and colon (feces) were similar (REF2). However, we agreed that the combination with cecal microbiota can better represent the gut microbiome influenced by diet and probiotic supplementation than fecal sample only.
REF1: Lee SW, Vineet S, Tatsuya U. Differences in fecal and cecal microbiota in C57BL/6J mice fed normal and high fat diet. Journal of Applied Biological Chemistry. 2021:64. 399-405.
REF2: Yan G, Li S, Wen Y, et al. Characteristics of intestinal microbiota in C57BL/6 mice with non-alcoholic fatty liver induced by high-fat diet. Front Microbiol. 2022: 1051200.
To prevent cross contamination, fresh feces were collected immediately after being excreted by each mouse in a sterile Eppendorf tube, frozen in liquid nitrogen and stored at -80°C before extraction.
We added the missing information in section 2.8.
Please, describe which bioinformatic tools were applied for sequenced data analysis and which database was used for the OTU identification?
The raw sequencing data or OTU table need to be provided as e.g. PubMed project.
Our response:
Thank you for your academic suggestion. The information has been added in section 2.8.
The raw sequencing data of OTU table was provided as the Excel film.
Line 169 Frozen faecal samples were extracted… Please, describe the method or include reference.
Line 177-184.
Have you used the same tools for microflora composition and function?
Our response:
Thank you for your professional suggestion. The method description of the frozen fecal samples extraction were revised and added in section 2.8. Meanwhile we used the same tools for the analyses of microflora composition and function.
- Results:
Figure 1A. Line 204.
You describe liver damage with lipid accumulation; however you have not performed lipid staining such as Oil Red O.
Our response:
Thank you for providing this comment. The HE staining could assess indirect evidence of lipid accumulation, and liver sections in the HFD showed significant fat accumulation from that in the chow diet in our previous studies (REF1-2). We also supplemented counted fat droplet numbers using Image J in figure 1.
REF1: Liu CY, Chang TC, Lin SH, Tsao CW. Is a Ketogenic Diet Superior to a High-Fat, High-Cholesterol Diet Regarding Testicular Function and Spermatogenesis? Front Nutr. 2022;9:805794.
REF2: Liu CY, Chen CC, Chiang LH, Yang BH, Chang TC, Tsao CW. Hirsutella sinensis intensifies testicular function and spermatogenesis in male mice with high-fat diet-induced obesity. J Chin Med Assoc. 2024;87:765-773.
Please, include the reference for H&E method that indicate the same. Show with arrows on the image fat accumulation. Indicate the scale bar for each image you included.
Our response:
Briefly, the paraffin section slides were deparaffinized and hydrated with xylene and serial alcohol solutions. Stained the sections with hematoxylin solution for 1 minutes, then wash with the running tap water for 2 minutes. Counterstaining of slides was done with Eosin for 1 minutes. Dehydrated the slides through alcohol, and clear in xylene, and mounted the slides with mounting medium.(REF 1)
REF 1: Ada T Feldman, Delia Wolfe. Tissue processing and hematoxylin and eosin staining. Methods Mol Biol. 2014;1180:31-43.
Thank you for your suggestion. We have added arrows showing hepatic steatosis and scale bar in each figure.
Figure 2. (A), Line 229.
I have doubts about depth of the brain tissue biopsy you show for each group. Please, indicate which mice brain matrice you used to dissect hippocampal area? Which cerebral hemisphere? Indicate the scale bar for each image you included. It looks like HFD image was taken with different magnification.
Our response:
We used the entire brain and sectioned at the largest cross-sectional region. The images were captured consistently form the left hemisphere. We have added the scale bar in each figure and changed the HFD image.
Lines 234-243. Please, specify in M&M how did you dissect out gastrocnemius muscle. From which side, left or right, gastrocnemius muscle was dissected and measured?
Our response:
The Morris water maze was conducted in the last week of probiotic treatment. The gastrocnemius muscle was dissected from the left side, and was consistent in all mice.
Figure 4. line 259. Indicate the scale bar for each image you included
Our response:
Thank you for your suggestion. We have added the scale bar in each figure.
Lines 281-301. The images authors show represent histological features of the small intestine (clear crypt-villus unit) but not cecum. Since authors have not specified the exact place of the biopsy collection, I assume that results might represent different parts from different groups and therefore cannot be compared.
Cecum is the part of large intestine and does not have villi. Please, check references for histological features of small and large intestines in mice.
https://currentprotocols.onlinelibrary.wiley.com/doi/full/10.1002/cpz1.1062
https://journals.sagepub.com/doi/full/10.1258/la.2009.009112
Figure 6 B. Are you presenting cecum with or without content? Please, specify in the text.
Our response:
You have raised an important question. We reconfirmed the tissue sections used in histological assessment and gut microbiota. The villus lengths and crypt depths were quantified using hematoxylin and eosin-stained distal ileal sections by Image J. We have rewritten the related sections in the article (Line) and added arrow in representative images of each group to indicate the colonic crypts (Figure 6A). The ileum was weighted without content.
Discussion:
Line 394: … middle-aged mice...
Should mice of 74 weeks of age be considered as old mice? Do you have reference?
Our response:
According to the Jackson Laboratory, McWain et al. and Wang et al. (REF1-3), mice of 74 weeks old correlate with humans of about 60-75 years old.
REF1: McWain MA, Pace RL, Nalan PA, Lester DB. Age-dependent effects of social isolation on mesolimbic dopamine release. Exp Brain Res. 2022;240(10):2803-2815.
REF3: Wang S, Lai X, Deng Y, Song Y. Correlation between mouse age and human age in anti-tumor research: Significance and method establishment. Life Sci. 2020; 242: 117242.
The obesity in middle-aged mice refers to mice from 37 weeks to 65 weeks old, the duration of high-fat diet-induced obesity.

Round 2
Reviewer 1 Report
Overall, the revision is satisfactory.
The authors did a good job!
Reviewer 2 Report
Clear improvement, acceptable for publication now
no comments
Reviewer 3 Report
Thank you for the revised version and provided references. In my opinion, the revised version is significantly improved and is suitable for publication.
No detail comments